# Validation of oxygen saturations measured in the community by emergency medical services as a marker of clinical deterioration in patients with confirmed COVID-19: a retrospective cohort study

Matthew Inada-Kim [ID],[1] Francis P Chmiel,[2] Michael Boniface,[2] Daniel Burns [ID],[2] Helen Pocock,[3,4] John Black [ID],[3,5] Charles Deakin[3,6]

For numbered affiliations see end of article.

**Correspondence to**
Dr Daniel Burns;
d.burns@soton.ac.uk

## ABSTRACT

**Objectives** To evaluate oxygen saturation and vital signs measured in the community by emergency medical services (EMS) as clinical markers of COVID-19-positive patient deterioration.

**Design** A retrospective data analysis.

**Setting** Patients were conveyed by EMS to two hospitals in Hampshire, UK, between 1 March 2020 and 31 July 2020.

**Participants** A total of 1080 patients aged ≥18 years with a COVID-19 diagnosis were conveyed by EMS to the hospital.

**Primary and secondary outcome measures** The primary study outcome was admission to the intensive care unit (ICU) within 30 days of conveyance, with a secondary outcome representing mortality within 30 days of conveyance. Receiver operating characteristic (ROC) analysis was performed to evaluate, in a retrospective fashion, the efficacy of different variables in predicting patient outcomes.

**Results** Vital signs measured by EMS staff at the first point of contact in the community correlated with patient 30-day ICU admission and mortality. Oxygen saturation was comparably predictive of 30-day ICU admission (area under ROC (AUROC) 0.753; 95% CI 0.668 to 0.826) to the National Early Warning Score 2 (AUROC 0.731; 95% CI 0.655 to 0.800), followed by temperature (AUROC 0.720; 95% CI 0.640 to 0.793) and respiration rate (AUROC 0.672; 95% CI 0.586 to 0.756).

**Conclusions** Initial oxygen saturation measurements (on air) for confirmed COVID-19 patients conveyed by EMS correlated with short-term patient outcomes, demonstrating an AUROC of 0.753 (95% CI 0.668 to 0.826) in predicting 30-day ICU admission. We found that the threshold of 93% oxygen saturation is prognostic of adverse events and of value for clinician decision-making with sensitivity (74.2% CI 0.642 to 0.840) and specificity (70.6% CI 0.678 to 0.734).

## STRENGTHS AND LIMITATIONS OF THIS STUDY

⇒ We used baseline community oxygen saturation measurements (on air) for COVID-19 patients conveyed by emergency medical services (EMS) to the hospital to evaluate the efficacy of these measurements as prognostic factors for short-term (30-day) intensive care unit admission and/or mortality.

⇒ We also assessed the prognostic value of the National Early Warning Score 2 and other vital signs measured by EMS to provide contrast with our oxygen saturation results.

⇒ The data are linked between EMS and hospital clinical records to enable our study.

⇒ The data have limitations: only patients conveyed by EMS were included, and the type of oxygen saturation measurement device for each patient was unknown.

## INTRODUCTION

SARS-CoV-2 is a highly transmissible and pathogenic coronavirus that causes COVID-19.[1] COVID-19 presents the biggest global healthcare challenge of our generation. As of February 2021, COVID-19-associated mortality stands at over 110 000 in the UK.[2] COVID-19 presents a number of challenges in identifying optimal management pathways, not only in terms of the clinical care itself but also in identifying the stage at which hospital admission is necessary. Traditional management pathways involving paramedic assessment and conveyance to the emergency department (ED) for further review have proven impractical, not only because of the large numbers of patients involved but also because of the need to minimise contact of COVID-19 patients with others. Most patients who become symptomatic do so in a home

environment where the majority will remain. In terms of optimising outcomes, there is a need to understand which symptoms and signs in this environment are prognostic indicators of potential deterioration. The national recommendation for the implementation of COVID virtual wards, recently announced by NHS England,[3] ushers in a novel approach to empowering patients by providing symptomatic, at-risk patients with a pulse oximeter and a toolkit for self-monitoring at home. It is hoped that this will enable earlier recognition of deterioration in COVID-19 patients and potentially improved outcomes.

In most cases of bacterial and non-COVID pneumonia, breathlessness appears relatively early in the disease and ahead of any significant hypoxia. The challenge with assessing COVID-19 severity is that asymptomatic hypoxia often precedes breathlessness, and by the time symptoms of breathlessness occur, patients have developed advanced disease and hypoxia may be significant.[4] The ability to detect this asymptomatic hypoxia before patients experience shortness of breath is critical for preventing respiratory involvement from progressing to a life-threatening state. The key is to be able to detect this initial drop in oxygen saturation levels so that patients infected with COVID-19 who begin to suffer from pulmonary complications in the community can be detected early and conveyed to the hospital for further treatment.[5] Although some studies have reported the relationship between oxygen saturation and outcome on presentation to the ED, we are not aware of any studies that have reported the relationship between oxygen saturation measured in the community by emergency medical services (EMS) and outcome. Patients who, on assessment, are severely hypoxic are clearly in need of emergency conveyance and hospital treatment, but by far the majority of patients with COVID-like symptoms seen and assessed by the EMS have relatively normal or near-normal oxygen saturations. These patients have generally not been conveyed and have been managed at home, but it has become apparent that even relatively minor derangements in oxygen saturations may be an early warning indicator for disease progression and the subsequent need for critical care. The use of oxygen saturation as an indicator of disease severity may therefore underestimate the risk of leaving patients at home after assessment by the EMS. National case fatality rates (ratio of deaths to total cases) have shown a strong inverse correlation between target oxygen saturation levels of 90%–98%,[6] suggesting that even mild derangements in oxygen saturation untreated can be detrimental to outcome.

Two small studies have suggested the utility of home oxygen monitoring for COVID-19 patients discharged from the hospital,[7 8] but no studies to our knowledge have used out-of-hospital oxygen saturation measurements as a trigger for initial hospital assessment. The purpose of this study, therefore, is to understand the prognostic significance of oxygen saturation when first measured by EMS clinicians. The understanding aims to inform escalation policies for safe and effective community-based triage and

self-monitoring at home by identifying a threshold where sensitivity and specificity are of clinical value. It is hoped that the approach will contribute to hospital admission avoidance, enable earlier recognition of deterioration in COVID-19 patients and potentially improve outcomes through early identification of those most at risk of disease progression. Using a pulse oximeter provides a way for patients to monitor disease progression through a simple measurement procedure, in contrast to the complexity of measurements required to calculate a National Early Warning Score 2 (NEWS2) score.

## METHODS

### Study design

We undertook a retrospective review of clinically confirmed COVID-19 patients accessing a regional UK ambulance service who were conveyed to the hospital and correlated their initial oxygen saturations measured at home with their in-hospital outcome. These were compared with the standard NEWS2 patient score, as used by all UK ambulance services, to identify the deteriorating patient.[9]

The cohort included adult patients (aged 18 years or older) initially assessed and conveyed by personnel from South Central Ambulance Service (SCAS) to the ED at one of the two hospitals within north Hampshire: Basingstoke and North Hampshire Hospital or the Royal Hampshire County Hospital (Winchester), at which the patients were subsequently admitted.

The standard care pathway included: (1) patients calling emergency (999) and urgent (111), where they are triaged using the NHS pathways telephone script (release 19); (2) attendance, assessment and monitoring by ambulance staff at the patient's home; (3) conveyance to the hospital for patients considered at high risk of deterioration and (4) admission to the hospital and escalation to the ICU for patients requiring critical care.

We analysed EMS conveyances occurring between 1 March 2020 and 31 July 2020 to determine suspect COVID-19 among conveyances at the initial time of contact by the call taker or EMS staff. Each patient record was reviewed for inclusion of at least one of the following four identifiers:

1. Those whom the EMS call taker had classified the call as 'COVID-respiratory distress'.
2. Those where the patient clinical record (PCR) listed the 'presenting complaint' as 'suspected COVID-19'.
3. Those where the PCR-free text for the 'presenting complaint' contained the word 'COVID'.
4. Those where the PCR narrative in the free text field summarising the symptoms and their details completed by the paramedic contained the word 'COVID'.

Conveyances from these suspected COVID-19 patients were then linked to their subsequent hospital attendance. Of the suspect cases, we then identified confirmed COVID-19 cases by selecting only those with a confirmed diagnosis in their discharge summary (ie, the presence

of a U07.1 or U07.2 ICD10 code). These confirmed COVID-19 cases made up our study cohort.

Seventeen patients did not have initial oxygen saturations recorded on air (but did have oxygen saturations recorded on oxygen) and were excluded from the data analysis. If this was because they were so obviously hypoxic clinically that EMS staff immediately administered oxygen without an initial reading on air (or were constantly on home oxygen treatment), the ability of oxygen saturations to indicate risk of deterioration is likely to have been underestimated in this study.

All patients in known palliative care pathways were excluded from data analysis because their care did not follow standard care pathways.

### Study setting
SCAS is a provider of emergency care in the counties of Hampshire, Berkshire, Buckinghamshire and Oxfordshire and covers a total of 3554 sq. miles ($9205 \text{ km}^2$). The service receives approximately 500 000 emergency and urgent calls annually. SCAS covers a residential population of approximately 4 million inhabitants in a mix of urban and rural areas. The north Hampshire region forms part of the area covered by SCAS and comprises a residential population of approximately 306 000.[10]

### Data collection
The initial oxygen saturation reading on air recorded by the attending EMS staff (prior to any exercise or step test) and the NEWS2 score of patients fulfilling the inclusion criteria were collected from the EMS PCR. (NEWS2 score is calculated using the following seven variables: systolic blood pressure, heart rate, respiratory rate, temperature, oxygen saturation, supplemental oxygen administration and level of consciousness: https://www.england.nhs.uk/ourwork/clinical-policy/sepsis/nationalearlywarning score).

Patient outcome was obtained by linking the SCAS and hospital clinical records by their NHS number. The primary outcome of our study was ICU admission within 30 days of conveyance, and the secondary outcomes were mortality and a combined outcome (ICU admission and/or mortality) within 30 days of conveyance.

### Data analysis
Analysis was performed in Python V.3.7.2, primarily making use of the statsmodels library. CIs on observed mortality rates were estimated using the Wilson score interval. Where relevant, the significance of the difference between two observed adverse outcome rates was tested using a two-population proportions z-test with the null hypothesis that the two-population proportions are equal.

To evaluate how predictive individual variables (eg, oxygen saturation) and combinations of variables (eg, oxygen saturation with age) were of 30-day adverse outcomes, we performed receiving operator characteristics (ROC) curve analysis. In the univariate analysis,

we performed a complete case analysis (removing any patient with an incomplete record of vital signs) and assumed a patient's adverse outcome risk is a linear function of the respective variable (where negative or positive correlation with outcome is assessed by clinical judgement) and calculated the ROC curve corresponding to whether this variable alone was used to predict a patient's risk of an adverse outcome. We present both the sensitivity and specificity of the area under the ROC (AUROC) curve. The AUROC provides an estimate of the degree to which the predictor can discern whether a patient has an adverse outcome within 30 days of conveyance or not; it can take values between 0.5 and 1. An AUROC of 0.5 corresponds to randomly guessing which patients will have an adverse outcome within 30 days and an AUROC of 1 corresponds to a perfect classifier; it can predict, without error, who will have an adverse outcome within 30 days of conveyance. CIs were estimated by performing 1000 bootstrapping (sampling with replacement) iterations on the available data, calculating the AUROC on each of the samples and then calculating the relevant percentiles.

### Patient and public involvement
This research was done without patient involvement. Patients were not invited to comment on the study design and were not consulted to develop patient-relevant outcomes or interpret the results. Patients were not invited to contribute to the writing or editing of this document for readability or accuracy.

## RESULTS
A total of 19 868 patients were assessed at home and subsequently conveyed by EMS to North Hampshire hospitals during the study period. The details of cohort selection are shown in figure 1. The call handler or EMS staff identified 2257 suspect COVID-19 cases and of these, we identified 1209 adults as having a confirmed diagnosis of COVID-19 (U07.1 or U07.2 coded in the patient discharge summary). Of the 1209 confirmed cases, we removed persons under palliative care (112 patients) and those with no initial oxygen saturation measurement on air recorded (17 patients). Overall, this left us with 1080 confirmed COVID-19 patient records, all of whom had initial oxygen saturation measurements on air. Of these 1080, the complete records of vital signs were recorded at home by paramedics for 892 of the patients. The summary of the final patient cohort with respect to demographics, comorbidities and the presence of vital sign measurements is given in table 1. In our following discussions, we make use of all 1080 patients, with the exception of our univariate analyses, where we perform a complete case analysis and only use the 892 complete records.

Oxygen saturation was found to correlate with adverse outcomes (figure 2A), with lower initial oxygen saturation readings being associated with a higher mortality rate. In figure 2A, we display the correlation between the

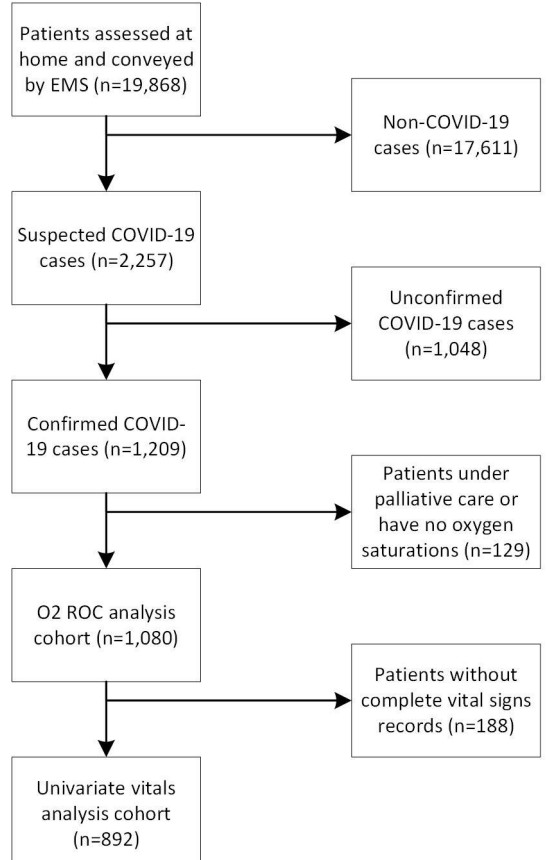

**Figure 1** The cohort selection of the emergency medical serivice patients.

observed 30-day adverse outcome rates and initial oxygen saturation in detail, which displays a correlation to all outcomes. In table 2, we display the breakdown of our retrospective ROC analysis for using measured oxygen saturation as a binary triage tool (ie, hospitalise or not) for different cut-offs (rows in table 2). While the sensitivity versus specificity trade-off needs to be determined by the clinical context, this demonstrates that oxygen saturation is moderately discriminative for several cut-offs. For example, for a cut-off of 94% or below, the sensitivity is 0.742 (95% CI 0.642 to 0.734) and the specificity is 0.706 (95% CI 0.678 to 0.734). Finally, we present comparisons of the results of ROC analysis for different variables measured in the community by EMS (table 3). Across the three presented outcomes (30-day ICU admission, mortality and combined outcomes), correlations between variables and outcomes are broadly similar, with measured oxygen saturations and the NEWS2 score being the two most predictive of outcome. The notable differences are for the measured temperature, which is moderately predictive of ICU admission (AUROC 0.720; 95% CI 0.640 to 0.793) but only weakly predictive of mortality (AUROC 0.597; 95% CI 0.523 to 0.678) and for patient age, which is strongly positively correlated to mortality but displays a negative correlation to ICU admission (figure 2B).

## DISCUSSION

Community assessment of patients with COVID-19 symptoms using a single initial oxygen saturation on air measurement correlates with 30-day clinical outcomes. Qualitatively, the observed 30-day adverse outcome rate is approximately constant between oxygen saturation of 100% and 96% and then increases with decreasing oxygen saturation from 95% to 90%. Below 90%, the mortality risk remains high. Although the therapeutic target range for oxygen saturation in the UK is 94%–98%[11] and in the USA is 92%–96%,[12] this study suggests that patients at the lower end of this range are still at risk of deterioration in the context of COVID-like symptoms. For example, patients in our cohort presenting with oxygen saturations in the range of 92%–94%, values often regarded as within this normal range, had a significantly (p=0.025) higher risk of ICU admission within 30 days (5.9%) compared with those presenting with oxygen saturations greater than 95% (ICU admission rate 2.5%). Outside this 'normal' range, our analysis suggests even relatively small decreases in oxygen saturation are markers of increased risk of death or ICU admission and suggests that a lower threshold for hospital conveyance may be necessary for patients who traditionally would be considered to have only minor physiological derangement and otherwise have been left at home.

The sensitivity of home oxygen saturation measurements reflects the percentage of people correctly identified as having adverse outcomes. The sensitivity of this parameter for adverse outcomes decreased as oxygen saturation fell (table 2). An oxygen saturation of ≤90% was associated with a relatively low sensitivity of <0.5. The specificity of identifying an adverse outcome, an indirect measure of unnecessary conveyance to the hospital (but also including patients who survived and did not need ICU admissions), increased as oxygen saturations fell. However, it is important to ensure that patients at risk of deterioration are not missed and a degree of overtriage would be necessary to ensure that this is not the case. However, even oxygen saturations at the lower end of the normal range are associated with a risk of deterioration (sensitivity of 94% saturation of 0.713), and it, therefore, appears that oxygen saturation alone has significant limitations when it is within a normal range.

Although oxygen saturations as a risk factor for COVID-19 patients on presentation to the ED are widely reported,[13 14] the ability of oxygen saturations measured in the community to indicate disease severity and the need for hospital conveyance has not been widely reported, presumably because of the challenges in equipping patients with pulse oximeters prior to the onset of any illness. Several studies have used oxygen levels in patients presenting in the ED as an indicator of the need for hospital admission and others have used the opportunity to send ED patients not requiring admission home with a pulse oximeter for self-monitoring. Oxygen saturations on presentation to the ED have also been shown to be strongly associated with outcome. The strongest critical

**Table 1** Characteristics of COVID-19-positive patients stratified by outcome

| Variable | Outcome category | | |
| --- | --- | --- | --- |
| Outcome | No adverse event (n=955) | 30-day ICU admission (n=58) | 30-day mortality (n=78) |
| Age | | | |
| 18–49 | 159 (16.6%) | 11 (19%) | 1 (1.3%) |
| 50–59 | 132 (13.8%) | 16 (27.6%) | 2 (2.6%) |
| 60–69 | 119 (12.5%) | 17 (29.3%) | 9 (11.5%) |
| 70–79 | 209 (21.9%) | 9 (15.5%) | 16 (20.5%) |
| 80+ | 336 (35.2%) | 5 (8.6%) | 50 (64.1%) |
| Comorbidities | | | |
| Chronic obstructive pulmonary disorder | 33 (3.5%) | 0 (0%) | 6 (7.7%) |
| Dementia | 90 (9.4%) | 1 (1.7%) | 18 (23.1%) |
| Diabetes | 216 (22.6%) | 14 (24.1%) | 14 (17.9%) |
| Kidney disease | 7 (0.7%) | 1 (1.7%) | 3 (3.8%) |
| Chronic pain | 37 (3.9%) | 3 (5.2%) | 1 (1.3%) |
| Vital signs | | | |
| Heart rate present | 946 (99.1%) | 58 (100%) | 77 (98.7%) |
| Systolic blood pressure present | 869 (91%) | 51 (87.9%) | 71 (91%) |
| Respiratory rate present | 852 (89.2%) | 49 (84.5%) | 70 (89.7%) |
| Oxygen saturation (on air) present | 955 (100%) | 58 (100%) | 78 (100%) |
| Temperature present | 825 (86.4%) | 49 (84.5%) | 67 (85.9%) |
| ACVPU present | 849 (88.9%) | 50 (86.2%) | 67 (85.9%) |

Note that 11 patients experienced both ICU admission and mortality within 30 days. We only report on comorbidities that were present in the dataset as provided by the emergency medical services. Comorbidity presence was recorded for every patient in the study. Oxygen saturations were not missing for any patients, as those with missing values had been excluded (n=17). Overall, vital signs records were complete in 83% of cases.
ACVPU, alert, confused, responding to voice, responding to pain, unresponsive; ICU, intensive care unit.

illness risk has been shown to be admission oxygen saturation of <88% (OR 6.99).[14] Other studies have shown that even a relatively mildly deranged oxygen saturation of <92% is strongly associated with an increased risk of in-hospital mortality.[15] Conversely, an ED resting oxygen saturation of ≥92% as part of discharge criteria can achieve hospital readmission rates as low as 4.6%,[16] suggesting that it may be a safe threshold for discharge in symptomatic patients with mild disease after diagnostic workup.

Home oxygen saturation monitoring has been used for patients discharged from the hospital, either from the ED because their disease was not severe or from intensive care for convalescence. A small study of patients with COVID-19 discharged from an ED reported similar results to ours using subsequent home oxygen saturation monitoring. In these patients, resting home oxygen saturation of <92% was associated with an increased likelihood of rehospitalisation compared with oxygen saturation of ≥92% (relative risk 7, 95% CI 3.4 to 14.5, p<0.0001). Home oxygen saturation of <92% was also associated with an increased risk of ICU admission.[8]

Oxygen saturation is an integral variable in most critical illness tools. The association of prehospital oxygen saturation has been shown to be predictive of 2-day mortality[17] and has been used to identify COVID-19 patients requiring hospital admission.[18] NHS England has encouraged the use of the NEWS2 scoring system to identify patients at risk of deterioration. This uses weighted physiological variables of heart rate, systolic blood pressure, oxygen saturation (on air), respiratory rate, temperature and level of consciousness to produce a score that is correlated with the risk of deterioration, not only as a general illness score but specifically in patients with known COVID-19.[19] NEWS2 has been compared with a quick COVID Sensitivity Index (qCSI), a test that includes oxygen saturation, respiratory rate and oxygen flow rate to calculate a score between 1 and 12 and a risk level. The study concludes that NEWS2 is significantly better than qCSI, with an area under the curve of 0.779 and 0.750, respectively.[20] Furthermore, qCSI does not consider severity scores for readings of 93% and above, while qCSI pulse oximetry readings are the lowest reading recorded during the first 4 hours of patient encounter at the hospital rather than prior to admission. In our study, we were concerned with the ability of isolated oxygen saturations measured by EMS on attendance in comparison with NEWS2 in our cohort to identify patients at

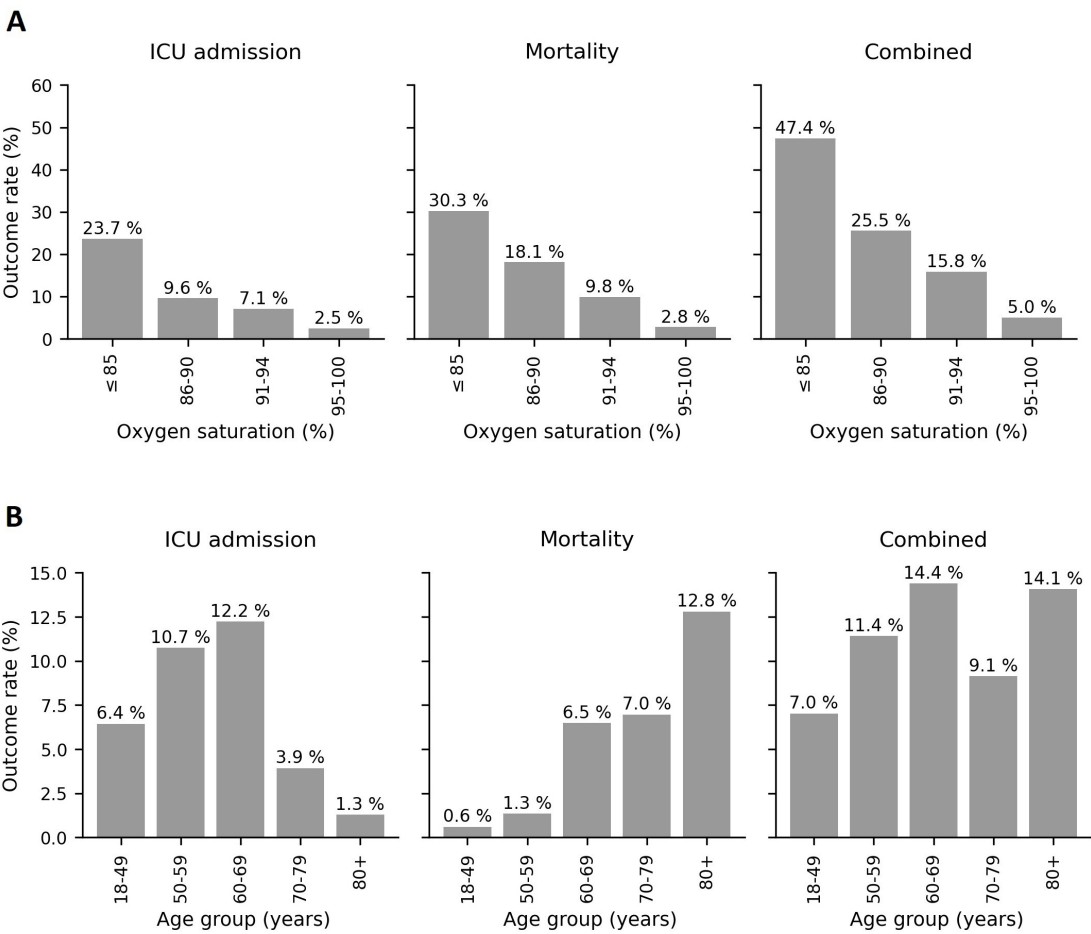

**Figure 2** The ICU admission, mortality and combined outcome rates as a function of (A) oxygen saturation percentage and (B) age group. ICU, intensive care unit.

risk of ICU admission (and mortality) within 30 days. Using ROC analysis, the AUROC for oxygen saturations at predicting ICU admission alone was 0.753 (95% CI 0.668 to 0.826) and for NEWS2 was 0.731 (95% CI 0.655 to 0.800). These results are consistent with a previous study using NEWS2 scores on hospital admission, which showed an AUROC of 0.822 (95% CI 0.690 to 0.953) to predict the risk of severe disease.[19] The lower observed AUROC of NEWS2 compared with oxygen saturations may be the result of the NEWS2 score incorporating physiological variables less predictive of COVID-19 outcomes than oxygen saturations, thereby reducing the discriminative ability of the score, or because it uses discretized oxygen saturations, which amounts to information loss. Additionally, we have not assessed the reporting compliance of the NEWS2 scores and this may have impacted the observed AUROCs. Interestingly, a recent review of 22 prognostic models showed that oxygen saturation in room air and patient age were strong predictors of deterioration and mortality among hospitalised adults with COVID-19, respectively, but no other variables added incremental value to these predictors.[18] We have shown the same for oxygen saturation as a univariate predictor in the prehospital setting and that predictive value does not increase with the addition of other physiological variables. The Pandemic Respiratory Infection Emergency System Triage (PRIEST) study using NEWS2, age, sex and performance status of patients in the ED predicted adverse outcomes with good discrimination in adults with suspected COVID-19.[21] The discriminatory ability of this more complex scoring system was similar to that demonstrated by simply measuring the oxygen saturations in the community and further reinforces the utility of home oxygen saturations as a simple marker, not only for use by the EMS but by members of the public equipped with home oximetry.

A number of remote home monitoring models for patients with suspected COVID-19 have been proposed, all of which aim to achieve early identification of deterioration for patients self-managing COVID-19 symptoms at home.[22] It would be expected that the utility of home monitoring would be improved by the ability to measure oxygen saturations, although not all models currently integrate this into their protocols. Our results show that resting oxygen saturations measured in patients with confirmed COVID-19 perform on par with the same measurements taken in the ED. They therefore suggest that the predictive value of oxygen saturations may be able to be effectively moved to an earlier stage in the disease process and measured while the patient is still

**Table 2** Evaluation of initial oxygen saturation measured by paramedics in COVID-19 patients in the community used as a binary classifier for predicting 30-day intensive care unit admission within 30 days of conveyance

| Oxygen saturation (on air) threshold (%) | Sensitivity (95% CI) | Specificity (95% CI) | Number of observations | Cumulative sum of number of observations |
|---|---|---|---|---|
| 85 | 0.294 (0.200 to 0.400) | 0.947 (0.933 to 0.962) | 8 | 76 |
| 86 | 0.316 (0.216 to 0.421) | 0.941 (0.927 to 0.955) | 8 | 84 |
| 87 | 0.320 (0.216 to 0.432) | 0.935 (0.920 to 0.950) | 6 | 90 |
| 88 | 0.370 (0.261 to 0.476) | 0.916 (0.899 to 0.933) | 23 | 113 |
| 89 | 0.413 (0.304 to 0.523) | 0.894 (0.874 to 0.913) | 25 | 138 |
| 90 | 0.512 (0.411 to 0.615) | 0.870 (0.849 to 0.890) | 32 | 170 |
| 91 | 0.590 (0.477 to 0.699) | 0.841 (0.823 to 0.867) | 31 | 201 |
| 92 | 0.655 (0.544 to 0.761) | 0.817 (0.796 to 0.841) | 33 | 234 |
| 93 | 0.706 (0.593 to 0.803) | 0.776 (0.751 to 0.801) | 45 | 279 |
| 94 | 0.742 (0.642 to 0.840) | 0.706 (0.678 to 0.734) | 74 | 353 |
| 95 | 0.808 (0.718 to 0.892) | 0.634 (0.605 to 0.662) | 76 | 429 |
| 96 | 0.848 (0.767 to 0.921) | 0.508 (0.477 to 0.538) | 129 | 558 |
| 97 | 0.898 (0.822 to 0.963) | 0.357 (0.330 to 0.386) | 156 | 714 |
| 98 | 0.911 (0.841 to 0.973) | 0.226 (0.201 to 0.254) | 132 | 846 |
| 99 | 0.961 (0.913 to 1) | 0.091 (0.075 to 0.109) | 139 | 985 |
| 100 | 1 | 0 | 95 | 1080 |

Each row denotes a different threshold for determining those at risk of an adverse outcome. We display the sensitivity and specificity for each threshold, equivalent to all possible intersections of the receiving operator curve, using thresholds between 85% and 100%. In total, 68 patients had an oxygen saturation of 84% or less (not shown). The column on the far right denotes the cumulative sum of the number of observations of the given oxygen saturation (row) or below. For example, 76 patients had an oxygen saturation of 85% or less recorded (top row) and 429 patients had an oxygen saturation of 95% or less recorded. CIs are estimated by bootstrapping.

at home. Although initial home oxygen saturation may provide a useful marker of disease severity and the need for hospital conveyance, it is clear that it has limited sensitivity and may need to be interpreted as part of an overall assessment of the patient. Some authors have argued that pulse oximetry identified the need for hospitalisation when using a cut-off of 92%,[8] but based on our data (table 2), approximately one-third of patients with an adverse outcome would be missed using this threshold. We have demonstrated that even patients presenting with oxygen saturations of 92%–94%, which are values often regarded as within a normal range, have a higher mortality rate than those with oxygen saturations higher than 95%. Even when measured in the ED, baseline median oxygen saturation was as high as 95% in those with an adverse outcome, compared with 97% in those without.[21] It is clear that the relatively low sensitivity of oxygen saturation in those with mildly deranged values limits the utility of this parameter alone in assessing the risk of adverse outcomes.

This is a relatively small retrospective cohort study with concomitant limitations in sample size. The subjective

**Table 3** Ranked AUROCs calculated for isolated physiological variables and the composite NEWS2 score with each outcome

| Variable | AUROC (95% CI) | | |
|---|---|---|---|
| | ICU admission | Mortality | Combined |
| Oxygen saturation (on air) | 0.753 (0.668 to 0.826) | 0.778 (0.704 to 0.843) | 0.775 (0.727 to 0.829) |
| NEWS2 | 0.731 (0.655 to 0.800) | 0.768 (0.709 to 0.823) | 0.760 (0.708 to 0.807) |
| Respiration rate | 0.672 (0.586 to 0.756) | 0.668 (0.599 to 0.736) | 0.677 (0.618 to 0.738) |
| Temperature | 0.720 (0.640 to 0.793) | 0.597 (0.523 to 0.678) | 0.636 (0.69 to 0.700) |
| Systolic blood pressure | 0.634 (0.560 to 0.706) | 0.604 (0.529 to 0.680) | 0.626 (0.568 to 0.684) |
| Heart rate | 0.590 (0.506 to 0.672) | 0.558 (0.486 to 0.631) | 0.574 (0.514 to 0.633) |
| Age band | 0.670 (0.611 to 0.734) | 0.685 (0.626 to 0.738) | 0.557 (0.495 to 0.615) |

AUROCs were calculated using a complete case analysis with 892 patients in total. CIs are estimated by bootstrapping, with 95% CIs presented alongside the mean validation AUROC across samples.
AUROC, area under receiver operator curves; ICU, intensive care unit; NEWS2, National Early Warning Score 2.

nature of the paramedic classification of symptoms consistent with COVID-19 may have introduced some degree of bias into the patients included in the study, as may have the presence of known comorbidities. Our dataset did not include patients who were reviewed by EMS but not conveyed to the hospital, and this is arguably the most significant source of bias in our study. It is reasonable that for patients where a decision was made not to convey them, they were less likely to deteriorate and more likely to have normal vital signs. If this is the case, this would result in a reduction in the discriminative ability of recorded oxygen saturations. We did not specifically compare the outcome data of COVID and non-COVID patients with mildly deranged oxygen saturations. However, our data suggests that mild derangement in COVID patients is a significant risk factor for deterioration and this does not match the clinical progression witnessed in non-COVID patients. We acknowledge that for very low oxygen saturation levels, our results show poor clinical value and we believe this is due to other factors influencing escalation decisions that are not included in our dataset. Patients on palliative care pathways were also removed from the study cohort, but are likely to be more susceptible to deterioration from COVID, irrespective of any alternative care pathway.

With waves of COVID-19 regularly overwhelming EMS and hospital services, there is an urgent need to optimise the identification of patients at risk of deterioration. We undertook this research to ascertain the role simple physiological measures might have in informing clinical decision-making. While the results are hypothesis-forming (ie, it shows oxygen saturations are predictive of clinical outcomes within the care pathway studied in this manuscript), it has clinical utility as it helps inform decisions made by clinicians at the point of conveyance. This will enable more patients to be safely managed in the community and only referred to the hospital once their clinical symptoms and physiological signs suggest a risk of deterioration and the need for hospital care. This is particularly needed for the majority of patients who have mild to moderate symptoms where it is not clear if community or hospital management is appropriate. Home pulse oximetry is becoming relatively cheap and easily accessible for the public and may be a relatively cost-effective tool in the safe community management of these patients, perhaps focused on those with significant comorbidities who are at higher risk. The utility of remote monitoring systems (or the COVID virtual ward) has been an increasingly studied subject, and there is growing evidence that remote monitoring can facilitate more streamlined approaches to the delivery of patient care, especially in pulmonary disease.[7] The use of ICU admission as an endpoint identifies patients seen at home who go on to deteriorate and the correlation of home oxygen saturation with a risk of severe deterioration assists ambulance crews in identifying both those who should be conveyed to the hospital and those who can, with a reasonable degree of certainty, be safely left at home. Further prospective studies are required to understand the utility of home pulse oximetry, but this study suggests that it may have the potential to significantly contribute to the safe and appropriate management of these patients in the community, with timely referral to the hospital when indicated.

## CONCLUSIONS

We have demonstrated that even relatively minor derangements in peripheral oxygen saturation are an early warning of potential deterioration in confirmed COVID-19 patients conveyed by EMS to the hospital and oxygen saturation would appear to have the potential to be a key physiological variable that, together with other clinical signs and clinical risk factors, may be able to identify patients at risk of deterioration.

**Author affiliations**
[1]Department of Acute Medicine, Hampshire Hospitals NHS Foundation Trust, Winchester, UK
[2]School of Electronics and Computer Science, University of Southampton, Southampton, UK
[3]South Central Ambulance Service NHS Foundation Trust, Otterbourne, UK
[4]Warwick Clinical Trials Unit, University of Warwick, Coventry, UK
[5]Emergency Department, Oxford University Hospitals NHS Foundation Trust, Oxford, UK
[6]Southampton Respiratory Biomedical Research Unit, University Hospital Southampton NHS Foundation Trust, Southampton, UK

**Acknowledgements** We thank Simon Mortimore and Philip King from South Central Ambulance Service and Zoe Cameron from Hampshire Hospitals NHS Foundation Trust for their assistance in data extraction and analysis.

**Contributors** MI-K, MB, JB and CD led and conceptualised the study. MI-K is the guarantor and led the overall study, with MB leading at UoS and CD at SCAS. FPC and DB performed the data analysis with support and guidance from all authors. MI-K, HP and JB performed the data extraction. MB led the data governance. CD and HP provided clinical insight. MI-K, CD, HP and FPC wrote the first draft of the manuscript. All authors discussed the results and contributed to subsequent drafts of the manuscript. DB prepared the final manuscript for submission.

**Funding** This report includes independent research funded by the National Institute for Health Research Applied Research Collaboration Wessex. The views expressed in this publication are those of the author(s) and not necessarily those of the National Institute for Health Research or the Department of Health and Social Care.

**Competing interests** MI-K is the National Clinical Lead Deterioration and National Specialist Advisor Sepsis, NHS England and NHS Improvement.

**Patient and public involvement** Patients and/or the public were not involved in the design, or conduct, or reporting, or dissemination plans of this research.

**Patient consent for publication** Not applicable.

**Ethics approval** Regulatory and ethical approval for the study were provided by the Health Research Authority (REC reference 20/HRA/5445) and by the University of Southampton Ethics Committee (REF ERGO/61242). NHS England and NHS Improvement have been given legal notice by the Secretary of State for Health and Social Care to support the processing and sharing of information to help the COVID-19 response under Health Service Control of Patient Information Regulations 2002 (COPI). This is to ensure that confidential patient information can be used and shared appropriately and lawfully for purposes related to the COVID-19 response. Data were extracted from medical records by clinicians providing care for the patients and an anonymised extract of the data were provided to the team at the University of Southampton.

**Provenance and peer review** Not commissioned; externally peer reviewed.

**ORCID iDs**
Matthew Inada-Kim http://orcid.org/0000-0001-6026-2246
Daniel Burns http://orcid.org/0000-0001-6976-1068
John Black http://orcid.org/0000-0002-1167-1550

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
