## [Reviewer comments · BMJ Open]

ARTICLE DETAILS

TITLE (PROVISIONAL)	Validation of oxygen saturations measured in the community by emergency medical services as a marker of clinical deterioration in patients with confirmed COVID-19 - a retrospective cohort study
AUTHORS	Inada-Kim, Matthew; Burns, Daniel; Chmiel, Francis P.; Boniface, Michael; Pocock, Helen; Black, John; Deakin, Charles

VERSION 1 – REVIEW

REVIEWER	Baek, Moon Seong Chung Ang University Hospital
REVIEW RETURNED	20-Sep-2022

GENERAL COMMENTS	Thank you for the opportunity to review this paper. Authors sought to investigate whether initial oxygen saturation by EMS staff was a better predictor of 30-day ICU admission compared to other physiologic parameters in COVID-19 patients. However, there are some concerns in the study design and result analysis. 1. The major limitation of this study is not presented the detailed patients flowchart and baseline characteristics. Outcomes such as 30-day ICU admission or mortality could be affected by patients' diagnosis (eg ARDS or sepsis), age, and commodities. Furthermore, detailed explanation of the reason for being contacted by EMS is needed.2. In this EMS setting, better triage of COVID-19 patients using oxygen saturation may be more appropriate than predicting 30-day mortality or ICU admission. If the oxygen saturation is low, it is likely to be hospitalized immediately in the ICU because it is severe in itself. Patients with normal saturation will be admitted to the general ward and then to the ICU if worsening. Therefore, it does not make sense that a measure of saturation itself would predict 30-day ICU admission. Therefore, the authors should present the time from EMS contact to ICU admission according to saturation.3. Line 41: Oxygen saturation is not the best predictor. The values of AUROCs of oxygen saturation and NEWS2 are not seem to be significantly different. If the authors want to argue that AUROC of oxygen saturation is better than those of NEWS2, please present the AUROC comparison by DeLong et al. (1988).4. Lines 47 (small deflection in oxygen saturation~): In the Abstract, the results did not revealed the evidence.
--

REVIEWER	Martín-Rodríguez, Francisco Universidad de Valladolid
REVIEW RETURNED	24-Oct-2022

GENERAL COMMENTS	The paper presented here is well written, well argued and with a clear methodology. It is to be appreciated that the authors of the paper "Validation of oxygen saturations measured in the community by emergency medical services as a marker of clinical deterioration in patients with confirmed COVID-19" have made this effort. However, I have some minor considerations: Pag 5 line 68, although it is obvious define the first time what COVID is, and differentiate COVID from SARS-CoV-2. At the end of the introduction should be clearly stated the objective of the study, now it is not appreciated. Page 8 line 153, they state that they calculate NEWS, why not NEWS2, and how many people had O2 at home, because it is to be assumed that these basic patients, due to their own pathologies, must have significantly lower saturation numbers. Indicate in the methos section with which devices the measurements were made Include a flowchart in the results, so that the reader can understand how from approximately 20,000 cases, the number of cases is reduced to 1,200. I am enclosing a couple of references that you have not included in your analysis and that I believe could help to liven up the discussion: Association of Prehospital Oxygen Saturation to Inspired Oxygen Ratio With 1-, 2-, and 7-Day Mortality (DOI: 10.1001/jamanetworkopen.2021.5700) One-on-one comparison between qCSI and NEWS scores for mortality risk assessment in patients with COVID-19 (DOI: 10.1080/07853890.2022.2042590)
--

VERSION 1 – AUTHOR RESPONSE

Reviewer: 1

Dr. Moon Seong Baek, Chung Ang University Hospital Comments to the Author:

Dear Editor,

Thank you for the opportunity to review this paper.

Authors sought to investigate whether initial oxygen saturation by EMS staff was a better predictor of 30-day ICU admission compared to other physiologic parameters in COVID-19 patients. However, there are some concerns in the study design and result analysis.

1. The major limitation of this study is not presented the detailed patients flowchart...

Response: We have amended the study design text to include a new paragraph 2 describing the patient flow

"The pathway includes 1) Patients calling emergency (999) and urgent (111) where they are triaged using NHS Pathways telephone script (release 19), 2) Attendance, assessment and monitoring by

ambulance staff at the patient's home, 3) Conveyance to hospital for patients considered at high risk of deterioration 4) Admission to hospital and escalation to ICU for patients requiring critical care. "

and baseline characteristics. Outcomes such as 30-day ICU admission or mortality could be affected by patients' diagnosis (eg ARDS or sepsis), age, and commodities.

Response: Table 1 now included showing distribution of age, available comorbidity information and vital signs measurement completeness.

Furthermore, detailed explanation of the reason for being contacted by EMS is needed.

Response: Patient's contacted EMS themselves according to the normal emergency care pathway. This is now described in 2nd paragraph of Study Design

2. In this EMS setting, better triage of COVID-19 patients using oxygen saturation may be more appropriate than predicting 30-day mortality or ICU admission.

Response: The study was conducted in 2020 in the context of emerging pathways for community-based triage assessment (hospital admissions avoidance) and patient reported monitoring of oxygen sats remotely from home through virtual wards. We state the pathway changes referring to NHS England "novel approach of empowering patients through providing symptomatic, at risk patients a pulse oximeter and a toolkit for self-monitoring at home. In this context patients were escalated to community virtual wards and/or hospital depending on oxygen saturation, symptoms and baseline risk factors. The purpose of the study was to provide early evidence for the oxygen saturation threshold.

If the oxygen saturation is low, it is likely to be hospitalized immediately in the ICU because it is severe in itself. Patients with normal saturation will be admitted to the general ward and then to the ICU if worsening. Therefore, it does not make sense that a measure of saturation itself would predict 30-day ICU admission. Therefore, the authors should present the time from EMS contact to ICU admission according to saturation.

Response: The purpose of the study has been elaborated in the final paragraph of the introduction section. A goal was to understand prognostic value of SpO₂ measurements in relation to adverse events and to try to identify a threshold where the sensitivity and specificity are of clinical value. We acknowledge that for very low SpO₂ levels our results show poor clinical value and we believe this is due to other factors influencing escalation decisions that are not included in our dataset. This has also been added to the discussion section.

3. Line 41: Oxygen saturation is not the best predictor. The values of AUROCs of oxygen saturation and NEWS2 are not seem to be significantly different.

If the authors want to argue that AUROC of oxygen saturation is better than those of NEWS2, please present the AUROC comparison by DeLong et al. (1988).

Response: We agree that Oxygen saturation offers similar predictive value to NEWS2. We were not intending to argue that oxygen sats were better than NEWS2. We state the need for "Understanding the prognostic implications of oxygen saturation when first measured by EMS clinicians would enable safe and effective triage and potentially improve outcome through early identification of those most at risk of disease progression.". We show comparable prognostic value, however, measuring oxygen saturation in the community is simpler to NEWS2. This is important for patient reported measurements required by remote monitoring on virtual wards.

We have adjusted the results to read “Oxygen saturation was comparably predictive of 30-day ICU admission (AUROC 0.753 (95 % CI: 0.668-0.826)) to the NEWS2 score (AUROC 0.731 (95 % CI: 0.655-0.800)), followed by temperature (AUROC 0.720 (95 % CI: 0.640-0.793)), and respiration rate (AUROC 0.672 (95 % CI: 0.586-0.756)).”

We have updated the discussion to highlight the need for simple but predictive patient reported measurements

4. Lines 47 (small deflection in oxygen saturation~): In the Abstract, the results did not revealed the evidence.

Response: We agree that deflection is the incorrect term. We have updated the conclusion to show that a threshold of 93% SpO₂ is prognostic of adverse events and of value for clinician decision making with sensitivity (74.2 % CI 0.642-0.840) and specificity (70.6 % CI 0.678-0.734)

Reviewer: 2

Dr. Francisco Martín-Rodríguez, Universidad de Valladolid Comments to the Author:

The paper presented here is well written, well argued and with a clear methodology. It is to be appreciated that the authors of the paper "Validation of oxygen saturations measured in the community by emergency medical services as a marker of clinical deterioration in patients with confirmed COVID-19" have made this effort. However, I have some minor considerations: Pag 5 line 68, although it is obvious define the first time what COVID is, and differentiate COVID from SARS-CoV-2.

Response: We have now defined SARS-CoV-2 and relationship to COVID-19 in the first sentence of the introduction with a new reference.

“Severe acute respiratory syndrome coronavirus 2 (SARS-CoV-2) is a highly transmissible and pathogenic coronavirus that causes coronavirus disease 2019 (COVID-19)”

At the end of the introduction should be clearly stated the objective of the study, now it is not appreciated.

Response: We agree and have restructured the final paragraph of the introduction as follows:

“The purpose of this study therefore is to understand the prognostic significance of oxygen saturation when first measured by EMS clinicians. The understanding aims to inform policies for safe and effective community-based triage and self-monitoring at home. It is hoped that the approach will contribute to hospital admission avoidance, enable earlier recognition of deterioration in COVID-19 patients and potentially improve outcome through early identification of those most at risk of disease progression. Whilst using a pulse oximeter provides a way for patients to monitor disease progression through a simple measurement procedure in contrast to the complexity of measurements required to calculate a NEWS2 score.”

Page 8 line 153, they state that they calculate NEWS, why not NEWS2,

Response: On review of the manuscript, we cannot find any references to NEWS. Only NEWS2 is used within this study as per reference: Royal College of Physicians National Early Warning Score (NEWS) 2. London: RCP, 2017. www.rcplondon.ac.uk/projects/outputs/national-early-warning-score-news-2 [Accessed 19 Oct 2020].

and how many people had O₂ at home, because it is to be assumed that these basic patients, due to their own pathologies, must have significantly lower saturation numbers.

Response: We agree that O2 at home would impact the results and is why we removed patients on O2 from the cohort. We state that all measurements were taken “on air”. The discussion section includes the statement

“Seventeen patients did not have initial oxygen saturations recorded on air (but did have oxygen saturations recorded on oxygen) and were excluded from the data analysis. If this was because they were so obviously hypoxic clinically that EMS staff immediately administered oxygen without an initial reading on air (or were constantly on home oxygen treatment), the ability of oxygen saturations to indicate risk of deterioration is likely to have been underestimated in this study.”

This paragraph has now been moved from the discussion to the study design to clearly indicate that patients on O2 were excluded from the cohort.

Indicate in the methods section with which devices the measurements were made

Response: The type of devices was not available in the data set. This is a limitation of the study and added to the list of limitations and in the discussion.

Include a flowchart in the results, so that the reader can understand how from approximately 20,000 cases, the number of cases is reduced to 1,200.

Response: We now include a flowchart diagram detailing cohort selection.

I am enclosing a couple of references that you have not included in your analysis and that I believe could help to liven up the discussion:

Association of Prehospital Oxygen Saturation to Inspired Oxygen Ratio With 1-, 2-, and 7-Day Mortality (DOI: 10.1001/jamanetworkopen.2021.5700)

Response: We have added reference has been added to the discussion sections.

One-on-one comparison between qCSI and NEWS scores for mortality risk assessment in patients with COVID-19 (DOI: 10.1080/07853890.2022.2042590)

Response: We have added a statement in the discussion section citing the comparison with qCSI.